# Proteomic Analysis and Biochemical Characterization of the Nematocyst Extract of the Hydrozoan *Velella velella*

**DOI:** 10.3390/md22100468

**Published:** 2024-10-12

**Authors:** Eleonora Tassara, Ivan Mikšík, Petr Pompach, Gian Luigi Mariottini, Liang Xiao, Marco Giovine, Marina Pozzolini

**Affiliations:** 1Department of Earth, Environment and Life Sciences (DISTAV), University of Genova, Via Pastore 3, 16132 Genova, Italy; eleonora.tassara@edu.unige.it (E.T.); gian.luigi.mariottini@unige.it (G.L.M.); 2Department of Analytical Chemistry, Faculty of Chemical Technology, University of Pardubice, Studentska 573, 532 10 Pardubice, Czech Republic; imiksik@seznam.cz; 3Institute of Biotechnology, Czech Academy of Sciences, 252 50 Vestec, Czech Republic; petr.pompach@ibt.cas.cz; 4Faculty of Naval Medicine, Naval Medical University, Shanghai 200433, China; hormat830713@hotmail.com

**Keywords:** hydrozoan, venom, proteome, toxins

## Abstract

The venom contained within cnidarian nematocysts has a complex composition and holds significant potential for biotechnological applications. In this context, one of the most effective methods for studying nematocyst contents is the proteomic approach, which can detect even trace amounts of compounds while minimizing the need for large-scale animal collection, thus helping to preserve ecosystem integrity. This study aimed to provide a comprehensive proteomic and biochemical characterization of the crude nematocyst extract from the common hydrozoan *Velella velella*. Despite not being harmful to humans, the analysis of the crude venom extract from *V. velella* brought to the identification of 783 different proteins, categorized into structural components, enzymes, and potential toxins, revealing a qualitative composition of the venom similar to that of other more toxic cnidarians. Biochemical assays confirmed the presence of various active hydrolytic enzymes within the extract, including proteases, phospholipases, hyaluronidases, DNases, and chitinases. These findings pave the road for future studies involving the pharmacological applications of *Velella velella* venom components through recombinant production and functional testing.

## 1. Introduction

Cnidaria is an ancient and worldwide spread phylum of mostly aquatic animals, present in both seawater and freshwater [1,2]. More than 12,000 species of cnidarians have been officially described to date [World Register of Marine Species, https://www.marinespecies.org/index.php, accessed on 19 April 2024], distributed in six classes: Scyphozoa (jellyfish), Anthozoa (anemones, sea pens), Hydrozoa (hydras), Cubozoa (box jellyfish), Staurozoa (stalked jellyfish), and Myxozoa (parasitic organisms with terrestrial hosts) [3,4]. The common feature among all members of this phylum is the presence of nematocysts, specialized venom-containing organelles used for prey capture and defense [1,5].

The composition of Cnidarian venom varies across different classes and species. This venomous arsenal contains a wide range of compounds, both proteinaceous and non-proteinaceous [6]. The first category includes enzymes that disrupt extracellular matrices and cellular membranes through lipolytic and proteolytic activities [7], along with proteinase inhibitors that enhance the venom’s effect and spread [8]. Additionally, it contains non-enzymatic proteins and peptides with toxic activities, such as blocking voltage-gated channels [9] or forming pores in phospholipid bilayers [10]. The non-proteinaceous components can be represented by purines that act, for instance, as adenosine receptor antagonists [11], by biogenic amines (such as histamine and serotonin), platelet-activating factors, prostaglandins, leukotrienes [12] with inflammatory effects, by neurotoxic quaternary ammonium compounds [13], and by betaines [14,15]. In humans, exposure to cnidarian venoms can trigger a large spectrum of symptoms, spanning from localized acute skin inflammation to more serious outcomes, such as cardiac and respiratory failure, neurotoxic effects, and, in extreme cases, even death occurring within minutes post-sting [16,17,18]. This represents a potential risk to human activities, especially in case of sudden jellyfish outbreaks occurrence, which frequency has increased since the last decades [17]. On the other hand, cnidarian venoms have led to ever-increasing attention towards their biotechnological and pharmacological potential: in fact, both crude and purified nematocyst extracts from many species have shown, among others, anti-cancer activity, anti-arrhythmic activity, insecticidal activity, radical-scavenging activity, anti-microbial activity, and immunomodulatory activity [18,19,20].

The great molecular complexity of cnidarian venom, also accounting how challenging it is to obtain enough fresh specimens, makes the proteomic approach one of the most effective methods for its study. The growing advancement of Next-Generation Sequencing (NGS) techniques and omics sciences has made the study of cnidarian venom and its pharmacological potential more accessible through proteomic approaches; this method bypasses time-consuming extraction processes and helps preserve the integrity of ecosystems, as it avoids the withdrawal of many animals from their habitat. Proteomic approaches have been successfully utilized to purify and characterize the venom components of many other venomous creatures besides jellyfish, such as snakes, spiders, scorpions, and cone snails [21,22,23,24]. The initial application of high-throughput mass spectrometry to characterize jellyfish venom identified the putative toxins extracted from the nematocysts of *Olindias sambaquiensis* [25], which was followed by numerous other studies [26,27,28]. When characterizing venoms, relying on peptide mass fingerprinting with databases primarily composed of protein sequences from model organisms can be limiting, particularly when the goal is to capture protein diversity. Utilizing species-specific transcriptomes addresses the absence of unique peptide sequences in these databases. Thus, combining transcriptomics and proteomics, the accuracy of venom profiling is significantly enhanced compared to using either method alone [29,30]. In this paper, we employed this approach to study the venom of the cnidarian *Velella velella*.

*Velella velella* (Linnaeus, 1758) is a blue-hued cosmopolitan hydrozoan present in open-ocean fauna [31]. Adult specimens are polymorphic colonies housed within a flattened oval chitinous pneumatophore. Hanging centrally beneath the sail is a large gastrozooid, involved in feeding and digestion [32], surrounded by hundreds of smaller gastro-gonozooids responsible for both digestive processes and reproduction [31,32]; these specialized polyps are enclosed by an additional ring of dactylozooids, which are involved in prey capture and contain the highest number of nematocysts, being mainly of the eurytele and stenotele types [17]. Each specimen contains about 3 × 10^5^ nematocysts; moreover, preliminary studies showed that crude venom extracted from these organelles displayed cytotoxic activity on L929 murine fibroblasts [17].

Building on previous findings that suggested a biotechnological potential for *V. velella*’s venom, this study aims to deliver a comprehensive proteomic characterization of the nematocyst extract from this species. By integrating high-throughput mass spectrometry with species-specific transcriptomic data, we intended to uncover the full range of protein diversity within the venom. Additionally, we performed enzymatic assays to experimentally validate the presence and activity of key enzymes identified in the proteomic analysis, further substantiating the functional components of this venom.

## 2. Results

### 2.1. Quantification of Total Proteins in the Crude Nematocyst Extract

The total protein content in the crude venom extract was quantified as described in Section 4 (see Figure 1), comparing a sample where the nematocysts suspension was subjected to a cycle of sonication (S) with a non-sonicated (NS) one. In the NS sample, the protein concentration resulted being 0.10 ± 0.04 mg/mL, while in the S sample, the concentration grew to 1.30 ± 0.26 mg/mL (see Appendix A). As reported by [17], the chosen methodology of extraction caused the discharge of the nematocyst’s thread and, consequently, the release of the nematocyst’s content. Therefore, the difference in protein concentration is mainly ascribable to the soluble protein fraction coming from the nematocysts after the treatment, as the structural proteins that constitute the nematocyst capsule are insoluble and were removed from the sample through centrifugation.

### 2.2. Transcriptome Assembly and Analysis

The results of the transcriptomic assembly performed with the Trinity-v2.14.0 software are presented in Appendix A, which reports the statistics of the assembled sequences’ length, their composition, and the quality and completeness assessment of the transcriptome using BUSCO and gVolante analyses. The assembly produced a total of 234,123 sequences. According to the quality and completeness analyses, 78.62% of the core genes queried were complete, 11.43% were fragmented, and 9.96% were missing. This transcriptome was subsequently used as a species-specific database for protein identification in the proteomic analysis.

### 2.3. Sodium Dodecyl Sulphate–PolyAcrylamide Gel Electrophoresis (SDS–PAGE) of the Crude Venom Extract

The protein content of the crude nematocyst extract from *Velella velella* was separated using Sodium Dodecyl Sulphate–PolyAcrylamide Gel Electrophoresis (SDS–PAGE). Due to the highly concentrated starting material, the gel appeared smeared, making it difficult to distinguish discrete protein bands. To ensure no information was lost, we excised the entire lanes rather than targeting individual bands. Each lane was divided into 11 distinct sections, as shown in Figure 1, and each section was prepared for further mass spectrometry analysis, allowing for a comprehensive analysis of all proteins present in the extract (see Section 4 for further details). The results of the mass spectrometry analysis are discussed in subsequent sections.

### 2.4. Characterization of Proteins Obtained from Proteomic Analysis

Mass spectrometry analysis identified 783 different proteins within the raw nematocyst extract from *Velella velella*, whose sequences are reported in Appendix A. The proteins were first separated using electrophoresis, and specific protein bands were excised from the gel, as shown in Figure 1. These bands were then adequately treated and subjected to mass spectrometry, which enabled the identification of the proteins present in the extract. The proteomics results were mapped against the assembled transcriptome, which served as a species-specific reference database. This approach allowed for more accurate identification and annotation of the proteins. After mapping, the identified proteins were annotated using BLAST and categorized into functional groups based on Gene Ontology (GO) terms (see Section 4 for details).

In a total of 783 proteins, 116 did not match against the queried BLAST database; they were thus subjected to a more in-depth analysis using modeling software (see the Section 4) and subsequently categorized as well based on modeling homology results. Table 1 presents the categories used to characterize proteins, and Table 2 shows the characterization of the previously unknown proteins, which were grouped into the same categories where applicable for ease of understanding. The categories “Toxins”, “Symbiont contaminants”, and “Fluorescent proteins” were added in Table 2.

A qualitative analysis of protein categories in the total extract was performed after annotating the unknown proteins and is shown in Figure 2, while the percentage of protein categories corresponding to each band excised from the SDS–PAGE is provided in Appendix A.

The most represented protein categories enlightened by the proteomic analysis of the crude nematocyst extract regard regulation or metabolic processes or constitute structural elements (see Figure 2). To obtain a better insight into the possible actual venom components, the categories that according to the literature [15,33,34,35] were more likely to display toxic activity were extrapolated from the total and reported in Table 3. Figure 3 shows the percentage distribution of potentially toxic elements in the extract. Notably, it was possible to identify 15 different classes of potentially toxic compounds, of which 50% are constituted by hydrolytic enzymes.

### 2.5. Evaluation of the Main Venom Hydrolytic Enzymes Activities in the Crude Nematocyst Extract

The activity of the main hydrolytic enzymes present in the crude venom extract of *Velella velella* was evaluated using in vitro assays. The results are presented in two sections: an evaluation of protease activity (Section 2.5.1) and an analysis of other hydrolytic enzymes (Section 2.5.2). Graphs and gel images are provided in Appendix A, while the data are summarized in Table 4 (for general protease activity with and without specific inhibitors) and Table 5 (for other enzymatic activities).

#### 2.5.1. Proteases Activity Evaluation

##### Generic Protease Activity Evaluation

The generic protease activity was quantified both on the crude venom extract alone and in the presence of specific protease inhibitors, as described in Section 4.7.1. The protease activity was quantified with respect to the absorbance of the positive control, containing 562 units of trypsin; considering that this amount of enzyme could digest an amount of substrate able to give a mean absorbance of 1.07 ± 0.01 (Standard Deviation, SD) at 440 nm with this assay, it is possible to say that the amount of crude nematocyst extract used for the assay contains a protease activity comparable to 306.66 ± 11.00 (SD) units of trypsin, equivalent to 1202.60 ± 47.00 (SD) protease units/mg of crude extract. When treated with specific inhibitors, in the assay this value goes down to 178.90 ± 10.90 (SD) protease units after phenylmethylsulfonyl fluoride (PMSF) treatment, equivalent to 701.60 ± 43.00 (SD) protease units/mg of crude extract, and to 189.93 ± 7.03 (SD) protease units after ethylenediamine tetraacetic acid (EDTA) treatment, equivalent to 744.80 ± 27.60 (SD) protease units/mg of crude extract, respectively (see Table 4). These results highlight the fact that the *V. velella* crude venom extract contains both serine-protease and metalloprotease activity in similar amounts.

##### Collagenase Activity Evaluation

To evaluate the presence of collagenase activity in the *Velella velella* sample, an in vitro assay was performed using the synthetic peptide Fa-LGPA as a substrate. The results are graphically presented in Appendix A and reported in Table 5. The positive control showed a consistent decrease in absorbance throughout the assay, indicating active collagenase-mediated cleavage of Fa-LGPA. In contrast, the absorbance values of the crude venom extract remained relatively stable, suggesting that no detectable collagenase activity was present in the nematocyst extract of *V. velella* under the tested conditions.

##### Zymographic Analysis of Active Proteases

To qualitatively identify the active proteases in the *Velella velella* nematocyst extract, a zymography analysis was performed using two different substrates: casein, to detect serine proteases, and gelatin, to detect metalloproteases (see Section 4). The results are presented in Appendix A. In both gels, proteolytic activity is visualized by clear bands against the blue background, indicating where proteases have digested the substrate. These results confirm, as previously demonstrated by the azocasein assay with specific inhibitors, that the crude venom extract contains both active serine proteases and metalloproteases. Specifically, proteases with a molecular weight of approximately 60 kDa were observed digesting casein, while proteases with a molecular weight of around 50 kDa were responsible for gelatin digestion.

#### 2.5.2. Evaluation of Other Enzymatic Activities

##### Hyaluronidase Activity Evaluation

As explained in the Section 4, a turbidimetric assay was conducted to quantify the potential hyaluronidase activity within the *V. velella* crude venom extract. The absorbance differences between the negative control and the *V. velella* venom extract are presented in Appendix A. The absorbance of the nematocyst extract sample is significantly lower than that of the negative control (*p* < 0.05), indicating that a portion of the hyaluronic acid in the solution was digested by the crude venom extract. By comparing the absorbance values to a standard curve of hyaluronic acid, it was determined that the crude venom extract contains approximately 2.00 ± 0.03 (SD) µg of hyaluronidase per mg of crude venom extract, as reported in Table 5.

##### Phospholipase A2 Activity Evaluation

The PLA2 in vitro turbidimetric assay was conducted using a principle similar to that of the collagenase and hyaluronidase assays, where enzymatic activity is indicated by a reduction in absorbance over time. Specifically, the absorbance value of the nematocyst extract sample at 740 nm decreased by an average of 71.53 ± 21.70% (SD) after incubation. This reduction corresponds to a mean activity of 12.11 ± 2.20 (SD) enzyme units per mg of nematocyst extract. The quantification of PLA2 activity in the crude venom extract is presented in Table 5.

##### DNase Activity Evaluation

The presence of DNase activity was assessed by treating *E. coli* genomic DNA with a known amount of nematocyst extract. Under our experimental conditions, 50 µg of *Velella velella* nematocyst extract contained sufficient DNase activity to completely degrade 250 ng of *E. coli* genomic DNA. The gel image showing the results is provided in Appendix A.

##### Chitinase Activity Evaluation

Chitinase activity in the crude venom extract was evaluated as described in Section 4.7.7. The activity was quantified by comparing the absorbance to a positive control containing 0.02 units of commercial chitinase. In this assay, 0.02 units of chitinase produced a mean absorbance of 1.37 ± 0.04 (SD) at 562 nm. Based on this, the samples containing crude nematocyst extract showed chitinase activity equivalent to 0.009 ± 0.001 (SD) units of commercial chitinase. This corresponds to a chitinase activity of 0.070 ± 0.001 (SD) units per mg of crude extract (see Table 5).

## 3. Discussion

This study presents the first comprehensive proteomic and biochemical characterization of the crude nematocyst extract from the hydrozoan *Velella velella*. Public SRA data were assembled to create a transcriptome database that is nearly 80% complete, making it a suitable species-specific resource for subsequent proteomic analysis. This high level of completeness enabled more accurate interpretation and identification of proteomics results compared to using generalized databases. Through mapping the proteomics data to the assembled transcriptome, a total of 783 unique proteins were identified. The most represented protein categories found in the extract include those involved in the regulation and modification of cellular processes, structural components, and extracellular matrix elements (see Figure 2). The predominance of these proteins is probably due to the methodology used for crude venom extraction: the sonication process, used for inducing nematocyst lysis, not only permits the release of the venom but also tends to disrupt the stinging organelle’s structural and functional elements. Consequently, many of the proteins identified are not directly associated with venom function but are more likely derived from the structure of the capsule and the thread. Indeed, among others, the typical short cysteine-rich fibrillary collagens constituting the capsule [36,37] were detected, as well as ion channels, transporters, and pumps which are thought to be essential in maintaining the internal osmotic pressure of the cyst [38,39,40]. However, a significant portion of the identified proteins actually turned out to be potential venom-related molecules according to the literature [33,34,35].

Considering the potentially toxic elements, hydrolases are the most represented group, comprising approximately 50% of the identified potential venom-related elements (see Figure 3). These hydrolases, half of which are proteases, were further evaluated through in vitro biochemical analyses to obtain better insight into their effective enzymatic activity within the crude venom extract.

Proteases are known to be an important fraction in the venom of many animals, such as snakes, spiders, and scorpions, as they play crucial roles in predation, digestion processes, and defense mechanisms [41]. The proteomic analysis of the *V. velella* crude venom extract (see Table 3) reveals the presence especially of metalloproteases (which, in venoms, are related to inflammatory and necrotic effects [42,43]) and, in lesser amounts, serine proteases (which have been linked to various physiological functions like platelet aggregation, fibrinolytic activity, enhancement of venom spreading in the target tissues, and post-translational modifications of other toxins [44]); the prevalence of these two categories of proteases has been observed in the venom of other cnidarians [45,46]. Notably, metalloproteases have been extensively studied in snake venoms and are often produced recombinantly to study their role in cancer progression and tissue remodeling, making them valuable for potential therapeutic use [42]. Serine proteases, known for their involvement in blood coagulation and fibrinolysis, are also studied in many venomous animals, including spiders and snakes, and have been recombinantly produced for use in anti-coagulant therapies [47]. In the crude venom extract of *Velella velella*, the azocasein assay revealed an overall protease activity almost comparable to that of purified trypsin (see Table 4); to better assess the contribution of each category to the total protease activity, the same assay was repeated in the presence of metalloprotease and serine protease-specific inhibitors (EDTA and PMSF, respectively). Notably, despite the transcriptomic analysis indicating a higher number of metalloproteases compared to serine proteases, the enzymatic activities of both categories were similar. This could be explained by different catalytic efficiencies or inactive metalloproteases within the extracted venom. Moreover, post-translational modifications or the presence of endogenous inhibitors [48] could also play a role in balancing the overall enzymatic activities of the two categories. Curiously, no collagenase activity was detected among the metalloproteases using our methods, despite collagenases being identified in other hydrozoans [49]. However, zymographic analysis revealed enzymes capable of digesting gelatine, a denatured form of collagen. This suggests the presence of other enzymes that, under physiological conditions, break down collagen in prey to facilitate gelatinase access to the already denatured molecule. The absence of detectable collagenase activity could be due to a low amount of collagenase in the crude venom, making the assay insufficiently sensitive. Alternatively, collagenase activity could be inhibited by quenching effects, which involve the reduction of enzyme activity due to interaction with other substances or intrinsic inhibitors present in the venom.

The other half of the hydrolases identified by proteomic analysis consists of a diverse assortment of enzymes confirmed to be active in the extract. These enzymes likely play crucial roles in the predation, feeding processes, and/or defense mechanisms of *V. velella*. For example, hyaluronidases, while not toxic themselves, act as spreading factors by enhancing the diffusion of toxins through the prey’s tissues [44]. Our methods detected hyaluronidase at a concentration of 2 µg/mL, corresponding to 1.2 µg of enzyme for each mg of total protein in the extract. This enzymatic activity aligns with the roles of phospholipases, which break down phospholipid bilayers, potentially enhancing the spreading of venom-related elements, as observed in many cnidarian extracts [50,51]. In particular, phospholipase D and Group XIIA secretory phospholipase A2, both found in many venoms, including that of *V. velella*, have been produced recombinantly and are valuable in lipid metabolism research. Group XIIA PLA2, in particular, has significant potential in anti-inflammatory drug development, as it can be used to study its mechanism of action and develop targeted inhibitors [52]. The presence of DNase activity, which is believed to assist in comprehensive cellular disruption, was also detected in venoms [15]. In the case of *V. velella*, an endonuclease activity capable of digesting prokaryotic genomic DNA was detected; however, our analysis was purely qualitative, and further studies will be necessary to quantify this enzyme. Nonetheless, the presence of this kind of hydrolase aligns with previous findings in *Physalia physalis* [53] and *Chrysaora quinquecirrha* [54], where endonucleases were purified and characterized. Finally, the presence of glycosidases—particularly chitinases—seems to be relevant considering the ecological niche of this hydrozoan, as a significant part of its diet consists of planktonic crustacean prey [55], whose exoskeletons are made of chitin. Chitinases have been recombinantly produced from bacterial and fungal sources, with applications in biopesticides and antifungal drug development [56]. These findings suggest that *V. velella* chitinases are worthy of further investigation to explore their potential for similar biotechnological applications.

The remaining 50% of potentially venom-related proteins in the *V. velella* nematocyst extract are non-enzymatic and were identified through proteomic analysis either as toxins or molecules capable of causing toxic effects in their targets. Of these, 30% are lectins. Lectins are carbohydrate-binding proteins, whose role in the venoms of many animals is to specifically target glycoproteins and glycolipids on the surface of cells, contributing to various physiological effects; more specifically, these molecules can facilitate the delivery of other harmful compounds by causing hemagglutination, thus impairing oxygen transport and causing tissue damage [57,58,59]. These molecules play roles in cell targeting, immune modulation, and hemagglutination, making them promising candidates for drug delivery systems and immunomodulatory therapies [60]. For instance, the Macrophage mannose receptor 1-like, found in *V. velella*, could be studied recombinantly produced to explore its structural or functional similarities to human receptors, potentially offering insights into novel mechanisms of immune modulation or therapeutic targeting in infections and cancers. Likewise interesting is the identification of homologs of toxins in the *V. velella* nematocyst extract, which have counterparts in many other venomous animals. These findings are significant because venoms exemplify evolutionary convergence, having evolved independently over 100 times across major animal groups, such as snakes, insects, arachnids, and jellyfish. These venoms have developed similar biochemical arsenals—especially neurotoxins, protease inhibitors, and pore-forming toxins—to immobilize prey and deter predators [61]. Notably, the *V. velella* nematocyst extract contains homologs of the Ly-6 neurotoxin, alpha-elapitoxin-Dpp2a, XaxB pore-forming toxin, kappa-stichotoxin-She3a, ShKT peptide ShKT-Ts1, and the Kunitz-type protease inhibitor-1 (see Table 3). The XaxB pore-forming toxin is a type of cytolysin that was first isolated from the bacterium *Xenorhabdus nematophila* [62]; however, sea anemones [63], spiders [64], and even some vertebrates [65] can also produce pore-forming toxins (PFTs), which share functional similarities with the bacterial ones. Regarding toxins distinctive of cnidarians, kappa-stichotoxin-She3a and ShK peptides are channel blockers found in the sea anemone *Stichodactyla helianthus*, whose toxic action can disrupt normal nervous and muscular function, leading to prey paralysis [66]; while these compounds are typical of sea anemones (Anthozoa), many homologues or similar domains in bigger proteins have been found in other cnidarians, such as *Hydra magnapapillata* and *Aurelia aurita* [51]. From a biotechnological point of view, neurotoxins such as Ly-6 and alpha-elapitoxin-Dpp2a target ion channels, leading to paralysis and neurological effects, making them interesting candidates for neurological drug development, particularly in targeting ion channels for pain management and neurological disorders [67]. Kunitz-type protease inhibitors, found in spiders, scorpions, and cnidarians, have been recombinantly produced and are highly relevant for developing cancer, thrombosis, and inflammation treatments [47].

Compared to other cnidarians, the venom of *Velella velella* displays both similarities and distinct features. Like *Physalia physalis* and *Chrysaora quinquecirrha*, *V. velella* venom contains a diverse array of enzymes, including proteases, phospholipases, and glycosidases [53,54]. Notably, *V. velella* venom also includes phospholipase D and Group XIIA secretory phospholipase A2, both of which are found in other cnidarians [50,51]. Additionally, homologous of neurotoxins and ion channel blockers in *V. velella* recall similarities with venom components found in sea anemones like *Stichodactyla helianthus* [66] and *Hydra magnipapillata* [51], which also target ion channels and disrupt nervous system function. However, the venom of *V. velella* seems to have a milder effect on humans compared to more potent cnidarian venoms, such as that of *Physalia physalis*, which is known to cause severe envenomation symptoms [53]. This suggests that while *V. velella* shares many venom components with other cnidarians, its lower toxicity may be due to differences in concentration or potency of the toxic elements; for instance, the mild effect of *V. velella*’s venom on humans may be due to the fact that, although it contains homologues of dangerous toxins, these compounds would need to be present in much higher concentrations to harm larger organisms. However, for *V. velella*’s purposes, the venom is likely potent enough to be lethal to its prey or deter attackers. For example, proteins containing ShK-like domains can block K+ channels, but while they may not affect humans at lower concentrations, they are likely sufficient to have a strong impact on smaller organisms [51,68]. Nonetheless, based on the current information, the precise physiological role of these substances in the organism remains unclear.

In conclusion, the presence of such compounds in an animal that does not provoke medically relevant effects on humans suggests a potential for biotechnological applications: molecules with specific targets but low toxicity are particularly interesting for drug discovery and development, as they are less likely to cause harmful side effects, crucial for developing safe therapeutic agents with a wider therapeutic window, especially for chronic use [69]. Further research should focus on selecting high-potential molecules from the nematocyst arsenal of *V. velella* for recombinant production and subsequent functional testing, which could reveal novel insights into their roles and contribute to the development of innovative therapeutic strategies. In particular, metalloproteases, phospholipase A2, chitinases, Kunitz-type protease inhibitors, and lectins stand out as the most promising compounds, given their established biotechnological applications and therapeutic potential in areas such as cancer, inflammation, thrombosis, and drug delivery systems.

## 4. Materials and Methods

### 4.1. Chemicals

Unless otherwise stated, all reagents were acquired from SIGMA-ALDRICH (Milan, Italy).

### 4.2. Velella velella Sampling

A small floating colony of *V. velella* was recovered in shallow coastal waters in the Gulf of Genoa area (Italy) in April 2023. The specimens were preserved in fresh seawater during their transfer to the laboratory, where they were promptly stored at −20 °C until future processing.

### 4.3. Crude Venom Extraction

The isolation of the nematocysts from the tissue was conducted as described in [17]. A group of about 20–25 specimens were soaked in a beaker containing artificial seawater and left to stir overnight at 4 °C. This process promotes tissue autolysis, leading to the subsequent release of intact nematocysts into the surrounding medium. Afterward, the sails were removed, and the whole content of the beaker was filtered through a cloth filter to remove chitin and tissue residues. The obtained suspension was then washed thrice by centrifuging for 10 min at 4000 rpm and resuspending the nematocyst pellet in clean seawater. After the last wash, the pellet was resuspended in 10 mL of a saline buffer (10 mM NaH_2_PO_4_ and 0.9% NaCl, pH 7.4). The nematocyst suspension was sonicated using a UP50H ultrasonic processor (Hielscher Ultrasonic, Teltow, Germany), subjecting the sample to 60 cycles of sonication of 10 s each and with a frequency of 30 kHz, always keeping it in ice. Finally, the suspension was centrifuged at 4 °C for 20 min at 18,000 rpm; the supernatant of the sonicated sample (S) was recovered and transferred into a clean tube. A non-sonicated sample (NS) was also centrifuged, and its supernatant was recovered to use it as a comparison and to confirm the breaking of the nematocysts and the release of the venom (see Figure 1). The crude extract thus obtained was then preserved at −20 °C until used for further experiments.

### 4.4. Quantification of Total Proteins in the Crude Nematocyst Extract

As cnidarian venoms are mainly constituted of proteinaceous compounds, such as enzymes, non-catalytic proteins, and peptides [15], to verify the release of the venom from the *V. velella* nematocysts, the total protein content in the soluble supernatant of both the non-sonicated (NS) and the sonicated (S) samples, obtained as described in Section 4.3, was assayed with the Bicinchoninic Acid Protein Assay kit, following the manufacturer’s instructions, in comparison to a BSA (Bovine Serum Albumin) standard curve. The absorbance of each sample was read at 562 nm using a Beckman spectrophotometer (DU 640, Beckman Coulter SpA, Milan, Italy). The procedure was carried out in triplicate.

### 4.5. Transcriptome Assembly and Putative Protein Database Construction

Sequence Read Archive (SRA) data available on NCBI (https://www.ncbi.nlm.nih.gov/, accessed on 6 June 2024; accession number: SRX4941821) were used to obtain a de novo assembled transcriptome for *V. velella*. A quality assessment of the raw reads was made with FastQC (v0.12.0) (https://www.bioinformatics.babraham.ac.uk/projects/fastqc/, accessed on 6 June 2024). Considering that a *Phred* score equal to 30 corresponds to a 99.9% base accuracy and a probability of 10^−3^ of incorrect base calling, the terminal bases resulted from the FastQC analysis with a Phred quality score < 30 were rejected using Trimmomatic (http://www.usadellab.org/cms/index.php?page=trimmomatic, accessed on 6 June 2024). Trimmed reads were re-screened using FastQC to check for residual adapters or primer sequences, which, if present, were discarded with Trimmomatic during a second round of processing. De novo assembling with clean data sets was performed with the Trinity software (https://github.com/trinityrnaseq/trinityrnaseq, accessed on 6 June 2024). The quality of the assembly was checked with Bowtie2 (https://bowtie-bio.sourceforge.net/bowtie2/index.shtml, accessed on 6 June 2024) and with the BUSCO tool (https://busco.ezlab.org/, accessed on 6 June 2024), together with a gVolante analysis (https://gvolante.riken.jp/, accessed on 6 June 2024). Finally, the transcript database was translated into proteins using the TransDecoder tool (https://github.com/TransDecoder/TransDecoder/wiki, last accessed 6 June 2024), where for each transcript, only the best open reading frame was retained; additionally, transcripts were filtered by length, with 20 amino acids set as the minimum, as some toxins can be very short. The translated transcript database was used as a reference database to search for protein identifications generated by the mass spectrometry (MS) analysis.

### 4.6. Proteomic Analysis

To obtain a complete protein profile of the nematocyst extract of *Velella velella*, a mass spectrometry-based proteomic analysis was performed.

#### 4.6.1. Sample Preparation

The sonicated extract (see Section 4.3) was further concentrated following the procedure described in [70]. In short, cold acetone was added to an aliquot of the extract in a 4:1 ratio; proteins were thus precipitated overnight at −20 °C. Samples were centrifuged for 10 min at 13,000 rpm and the supernatant was discarded, letting the protein pellet dry in uncapped tubes for 30 min and then resuspending it in an adequate volume of 10 mM NaH_2_PO_4_ and 0.9% NaCl, pH 7.4. The concentrated extract was subsequently used to perform two SDS–PAGE with a Mini-Protean 3 (Bio-Rad Laboratories, Hercules, CA, USA), following the method previously described in [71]. A 5 mg/mL solution of crude extract (value referred to the total protein content after acetone concentration) was mixed at a 3:1 (*v*/*v*) ratio with a 4× gel loading buffer (1 M Tris–HCl buffer, pH 6.8, 10% 2-mercaptoethanol, 40% glycerol, 0.2% bromophenol blue and 20% sodium dodecyl sulfate solution) and heated at 90 °C for 10 min. Consequently, 40 μL of the resulting solution (thus containing a total of 150 μg of proteins) were loaded into 14 wells of the two 12% polyacrylamide gels and run at 60 mA with constant amperage, using 10 μL of PageRuler™ Prestained Protein Ladder as standard. At the end of the run, gels were briefly rinsed with distilled water and fixed for 1 h in a fixing solution containing 10% (*v/v*) acetic acid and 40% (*v/v*) ethanol, then washed twice for 10 min with distilled water and stained overnight with a staining solution containing a 1:4 (*v*/*v*) *ratio* of methanol and colloidal Coomassie solution (0.1% (*w/v*) Coomassie Brilliant Blue G250, 2% (*w/v*) orthophosphoric acid, 10% (*w/v*) ammonium sulfate). Each one of these steps was conducted at room temperature. Finally, gels were destained with a 5% acetic acid solution. As the crude venom extract contains a mixture of proteins, to obtain an adequate concentration of each protein suitable for further mass spectrometry analysis, the 14 resulting lanes were precisely excised at identical points using a sterile scalpel. This process yielded 11 bands, each representing the combined slices from the same points across all 14 lanes. Figure 1 reports a single SDS–PAGE lane and the corresponding bands into which it was divided. Slices were cut into cubes, transferred into 1.5 mL tubes, and prepared for mass spectrometry analysis following the method reported in [72]; after reduction, alkylation, and destaining (see [72]) proteins within the gel pieces were digested overnight at 37 °C with a 13 ng/µL trypsin solution dissolved into 10 mM ammonium bicarbonate containing 10% *(v/v)* acetonitrile. Then, peptide digestion products were recovered by adding into each tube 100 µL of extraction buffer (1:2 5% *(v/v)* formic acid/acetonitrile) and incubating for 15 min at 37 °C in a shaker, further recovering the supernatant in a PCR tube. This step was repeated twice. Finally, samples were dried down in a vacuum centrifuge before mass spectrometry analysis.

#### 4.6.2. Mass Spectrometry Analysis

Samples digested by trypsin were analyzed using a liquid chromatography system Vanquish (Thermo Scientific, Waltham, MA, USA) connected to the timsToF SCP mass spectrometer equipped with Captive spray (Bruker Daltonics, Billerica, MA, USA). The mass spectrometer was operated in a positive data-dependent mode. One microliters of peptide mixture were injected by autosampler on the C18 trap column (Pepmap Neo C18 5 µm, 0.3 × 5 mm, Thermo Scientific). After trapping, peptides were eluted from the trap column and separated on a C18 column (Pepsep C18 150 × 0.15 mm, 1.5 µm, Bruker Daltonics) by a linear 35 min water-acetonitrile gradient from 5% (*v/v*) to 35% (*v/v*) acetonitrile at a flow rate of 1.5 µL/min. The trap and analytical columns were both heated to 50 °C. Parameters from the standard proteomics PASEF method were used to set timsTOF SCP. The target intensity per individual PASEF precursor was set to 20,000, and the intensity threshold was set to 1500. The scan range was set between 0.6 and 1.6 V s/cm^2^ with a ramp time of 100 ms. The number of PASEF MS/MS scans was 10. Precursor ions in the m/z range between 100 and 1700 with charge states ≥2+ and ≤6+ were selected for fragmentation. The active exclusion was enabled for 0.4 min. The raw data were processed by PeaksStudio 11.0 software (Bioinformatics Solutions, Waterloo, ON, Canada). The search parameters were set as follows: enzyme—trypsin (specific), carbamidomethylation as a fixed modification, oxidation of methionine, and acetylation of protein *N*-terminus as variable modifications. The MS outcome was searched against the *V. velella* putative protein database obtained by translating the assembled transcriptome (as described in Section 4.5).

#### 4.6.3. Protein Characterization

The data set resulting from mass spectrometry analysis was investigated using the Basic Local Alignment Search Tool (BLAST, https://blast.ncbi.nlm.nih.gov/Blast.cgi, accessed on 13 June 2024) against a UniProtKB/Swiss Invertebrate proteins database (https://www.uniprot.org/help/downloads, accessed on 13 June 2024), and the UniProtKB/Swiss-Prot Tox-Prot database [73] using default settings, considering an e-value ≤ 1 × 10^−5^. Then, proteins were functionally classified using Gene Ontology categories (Gene Ontology Resource, accessed on 13 June 2024). The proteins that remained uncharacterized after the BLAST analysis were manually identified using SMART (http://smart.embl-heidelberg.de/, accessed on 13 June 2024), Swiss Model (https://swissmodel.expasy.org/, accessed on 13 June 2024), and/or Phyre2 (http://www.sbg.bio.ic.ac.uk/~phyre2/html/page.cgi?id=index, accessed on 13 June 2024), by singularly subjecting the uncharacterized sequences to the above-mentioned prediction tools.

### 4.7. In Vitro Evaluation of the Main Crude Venom Hydrolytic Enzymes Activities

The presence of some classes of hydrolytic enzymes commonly found in the venom of cnidarians [15] was experimentally evaluated by some in vitro trials on the crude extract obtained from *V. velella*. More specifically, the activity of generic proteases (with and without specific protease inhibitors), collagenases, hyaluronidases, phospholipases, chitinases, and DNases was investigated.

#### 4.7.1. Generic Protease Activity

The generic protease activity was evaluated and quantified following the method described by [74], using azocasein as an enzymatic substrate. As an enzymatic source, 150 μL of 1.7 mg/mL crude venom was added to 600 μL of a solution, previously heated up to 37 °C, composed of 225 μL of reaction buffer (100 mM Tris-HCl, pH 8; 20 mM CaCl_2_) and 375 μL of 2% azocasein solution, pH 8. Positive and negative controls were also prepared in the same way using, respectively, 0.5 mg/mL trypsin and a thermally inactivated aliquot of the crude venom extract itself. For the blank, the enzymatic source was replaced by the same volume of reaction buffer. The samples were incubated at 37 °C for 30 min. After the incubation, 500 μL of each sample was transferred into clean tubes, and an equal volume of 110 mM trichloroacetic acid was added; then, they were centrifuged at 16,000 rpm for 10 min. The supernatant was recovered and added to 500 μL of 500 mM NaOH. Finally, samples were read against blank at a 440 nm wavelength using a Beckman spectrophotometer (DU 640, Beckman Coulter Spa, Milan, Italy). The procedure was carried out in triplicate.

To better determine what kind of proteases were contained in the sample, the same experiment was repeated after treating the crude venom extract with specific protease inhibitors, following the example of [75]. Ethylenediaminetetraacetic Acid (EDTA), which inhibits metalloproteases, or phenylmethylsulfonyl fluoride (PMSF), as a serin-proteases inhibitor, was added to the *V. velella* extract to reach a final concentration of 5 mM. In both cases, the extract was incubated with the specific inhibitor for 1 h at 4 °C before proceeding with the assay. The procedure was carried out in triplicate. Results were expressed in Units of enzymatic activity/mg of crude extract with respect to the activity of positive control.

#### 4.7.2. Collagenase Activity

The collagenase activity was measured with a turbidimetric assay as explained in [76], using the synthetic peptide 2-furanacryloyl-l-leucylglycyl-l-prolyl-l-alanine (Fa-LGPA) as an enzymatic substrate. A volume of 20 μL of 1.7 mg/mL crude venom extract was added to 40 μL of a 5 mM Fa-LGPA solution, prepared by dissolving the peptide in a reaction buffer (50 mM tricine, 10 mM CaCl_2_, 400 mM NaCl, pH 7.5). As a positive control, 20 μL of a 2 U/mL aqueous solution of commercial collagenase from *Clostridium histolyticum* was used. A negative control was also prepared by thermally inactivating an aliquot of the crude venom extract, specifically by treating it at 95 °C for 15 min. The samples were read at a wavelength of 345 nm every 15 s for 15 min, using a Beckman spectrophotometer (DU 640, Beckman Coulter SpA, Milan, Italy). The units of enzyme in the crude venom extract were calculated as follows:$$Units/mL\;enzyme=\frac{\left(\Delta A345/min\;Sample-\Delta A345/min\;Blank\;(0.06\right)}{\left(0.53)\;(0.02\right)}$$
where

ΔA345 = delta of absorbance between t_1_ and t_0_

0.06 = Total volume (mL) of the reaction mixture

0.53 = Millimolar extinction coefficient of Fa-LGPA at 345 nm

0.02 = Volume (mL) of enzyme solution/sample used

#### 4.7.3. Zymographic Analysis of Active Proteases

A zymography assay was conducted to detect the proteolytic activity of *V. velella* crude venom using casein and gelatin as substrates. This method, outlined in prior studies [27,77], was slightly modified. Gelatin (1.2 mg/mL) or casein (1.2 mg/mL) were copolymerized with 12% SDS–polyacrylamide gels. Samples were prepared under non-reducing conditions by dissolving them in a loading buffer with final concentrations of 0.8% *(w/v)* SDS, 4% *(v/v)* glycerol, 0.002% *(w/v)* bromophenol blue, and 25 mM Tris-HCl at pH 6.8. Each well was loaded with 30 μg of proteins, including both the positive control (protease from *Streptomyces griseus*) and the crude venom extract. Electrophoresis was carried out at 4 °C with a constant voltage of 200 V. At the end of the running, gels were rinsed with deionized water and washed twice with 2.5% *(v/v)* Triton-X for 30 min and rinsed again; then, it was reconditioned for 5 min with an incubation buffer (0.2 M NaCl, 50 mM Tris-Base, 5 mM CaCl_2_, 5 μM ZnCl_2_, pH 7.6). The development of the gel was carried out overnight in the same incubation buffer, at 37 °C. The gel was then rinsed and stained with 0.8% *(w/v)* Coomassie Blue—G250 in 45 mM HCl, putting it in a microwave oven for about 10 s at a power of 500 W. Before staining, the lanes containing the pre-stained molecular weight standards were separated from the rest of the gel to prevent their reduced visibility due to overall staining caused by the embedded protein substrates. After staining, the gel portions containing the standards (whose background thus remained clear) were reassembled with the rest of the gel for visualization.

Finally, the gel was destained with deionized water until clear zones of hydrolysis in the gel against the blue background showed gelatin proteolytic activity.

#### 4.7.4. Hyaluronidase Activity

The presence of hyaluronidases was evaluated using the method described in [78], with some modifications. A total of 100 µL of 1.7 mg/mL crude venom extract was mixed with 400 µL of a 0.5 mg/mL hyaluronic acid solution, prepared by dissolving hyaluronic acid in acetate buffer (0.2 M C_2_H_3_NaO_2_ in acetic acid, 0.15 M NaCl, pH 6.0). An aliquot of the crude venom extract was thermally inactivated by keeping it at 95 °C for 15 min and used as a negative control. The samples were confronted with a standard curve of commercial hyaluronidase from *Streptomyces hyalurolyticus*. The hyaluronic acid solution is substituted for the blank with the same volume of acetate buffer. All samples were incubated at 37 °C for 15 min; consequently, the reaction was stopped with 2 mL of cetyltrimethylammonium bromide (CTAB) solution (2.5% CTAB in 2% NaOH, pH 12). After 10 min at room temperature, samples were read at a 400 nm wavelength against blank with a Beckman spectrophotometer (DU 640, Beckman Coulter SpA, Milan, Italy). As the CTAB solution forms an insoluble product reacting with hyaluronic acid, the lower the absorbance value, the higher is the amount of hyaluronic acid digested. The experiment was performed in triplicate.

#### 4.7.5. Phospholipase A2 Activity

The presence of PLA2 in the crude venom extract from *V. velella* was investigated following the method of [79]. A solution of egg yolk substrate was created by dissolving the egg yolk in a 0.9% NaCl solution and adjusting the absorbance to 1 optical density (OD) at 740 nm, using a 20 mM Tris-HCl buffer (pH 7.4). A volume of 60 μL of 1.2 mg/mL crude venom extract was added to 250 μL of the egg yolk solution, and the absorbance was read at 740 nm (t_0_) with a Beckman spectrophotometer (DU 640, Beckman Coulter SpA, Milan, Italy). Then, the sample was incubated at 37 °C for 60 min and the absorbance was read again (t_1_). The procedure was carried out in triplicate, using a commercial phospholipase from *Apis mellifera* as a positive control and a thermally inactivated *V. velella* crude venom sample as a negative control. The unit activity of PLA2 activity is described as the amount of crude venom PLA2 which decreases the turbidity of the solution by 0.01 absorbance unit per minute at 740 nm. The unit activity of PLA2 was then calculated as the quantity of crude venom PLA2 that reduces the turbidity of the solution by 0.01 absorbance unit per minute at 740 nm.

#### 4.7.6. DNase Activity

The DNase activity in the nematocyst extract of *V. velella* was researched as described in [79]. A quantity of 50 μg of nematocyst extract (quantity referred to the total protein content) was incubated at 37 °C for 1 h together with 250 ng of genomic DNA from *Escherichia coli*. An untreated sample of DNA was used as a negative control. The assay mixtures were subjected to 0.8% agarose horizontal gel electrophoresis and its image was acquired with a ChemiDoc Imaging System (Bio-Rad, Milan, Italy).

#### 4.7.7. Chitinase Activity

The presence of chitinase activity was investigated using a combination of methods described by [80,81], with some modifications. A colloidal chitin solution was prepared by dissolving commercial crab chitin in ten volumes of 37% (*v*/*v*) HCl for 1 h at room temperature. Afterward, 1000 volumes of cold ethanol were slowly added, and the suspension was centrifuged at 8000× *g* for 20 min. The resulting pellet was collected and washed with distilled water until reaching a neutral pH. The assay reaction mixture consisted of equal volumes (50 µL each) of colloidal chitin solution, 50 mM sodium acetate buffer, pH 5.2, and *V. velella* nematocyst extract at a concentration of 2.4 mg/mL. For the positive control, a 1.2 mg/mL chitinase solution from *Streptomyces griseus* was used, while the negative control contained the same volume of sodium acetate buffer. Samples were then incubated at 37 °C for 1 h. Following incubation, 100 µL of 5% trichloroacetic acid were added to each sample to precipitate all proteins. The samples were centrifuged at 12,000× *g* for 10 min, the supernatant was recovered into clean tubes and the pH was adjusted to neutral. Finally, the reducing sugars produced from the chitin digestion were evaluated using a modified bicinchoninic acid assay [81] to measure the activity of the enzyme. The absorbances were read at 562 nm using an AMR-100 microplate reader. The procedure was carried out in triplicate and results were expressed in units of enzyme/mg of crude extract with respect to the activity of positive control.

## 5. Conclusions

This study presents the first comprehensive evaluation of the nematocyst extract of the hydrozoan *Velella velella*, utilizing a combination of transcriptomic, proteomic, and biochemical approaches. The analysis revealed a complex mixture of components with a qualitative composition similar to that of more toxic cnidarians, including both spreading factors and homologues of well-known toxins. However, unlike more dangerous species, contact with *V. velella* does not cause medically significant effects in humans. While many effective drugs are derived from highly toxic compounds, their extreme toxicity often complicates clinical trials and requires modifications to reduce harmful effects. In contrast, low- or non-toxic molecules, such as those found in *V. velella*, may require fewer adjustments, making them more promising candidates for drug development with minimal side effects. Additionally, the species-specific database developed in this study allows for the identification of novel compounds through in silico methods; these compounds can be produced with recombinant methods and functionally tested without the need for laborious purification processes, offering potential as bioactive compounds in pharmaceutical applications. Future work could further enhance the transcriptome assembly by incorporating alternative assembly software to explore greater diversity among transcript families [82], aiding in building on the foundation laid in this study and potentially uncovering additional novel bioactive compounds.

## Data Availability

The data presented in this study are available on request from the corresponding author.

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
