# Peer review of "Proteomic Analysis and Biochemical Characterization of the Nematocyst Extract of the Hydrozoan Velella velella"

_marinedrugs, 2024, doi:10.3390/md22100468_

Round 1
Reviewer 1 Report
Comments and Suggestions for Authors
Dear Authors,
Thank you for your good work. The results obtained can be compared with the proteomic contents of other species. Also, you can find a few small suggestions in the file.
Best

Comments on the Quality of English Language
Some terms may be edited for the sake of completeness of the article. For example, "toxin" and "venom" or "raw" and "crude". Whichever you prefer...
Author Response
Reviewer 1
Added phrases addressing R1’s comments were highlighted in yellow in the text.
Reviewer comments:
Dear Authors,
Thank you for your good work. The results obtained can be compared with the proteomic contents of other species. Also, you can find a few small suggestions in the file.
Best
Some terms may be edited for the sake of completeness of the article. For example, "toxin" and "venom" or "raw" and "crude". Whichever you prefer.
Response from the authors:
The authors thank the reviewer for the suggestions. We have revised the manuscript, opting to retain the word "crude" for consistency throughout the text. Additionally, we have included a new paragraph in the discussion to provide a more thorough comparison of the venom composition of Velella velella with that of other cnidarians.
Reviewer 2 Report
Comments and Suggestions for Authors
The authors present a massive amount of work describing the venom composition and function of venom from Velella velella. The authors assembled a previously published transcriptome and conducted a proteomic analysis of the venom. The authors then performed numerous experiments set to isolate and describe different enzymatic properties of the venom.
Major points:
The current description of the results of the transcriptome are very confusing. The authors do not present how many sequences that they assembled for toxin annotation or how many of those sequences matched with the proteome. The authors present the BUSCO scores of how many conserved genes were assembled with the trinity which help to understand the quality of the assembly but does not give any information to how the transcriptome was useful to the study.
- Could the authors provide more information to how the transcriptome was actually used and what information was pulled from this data?
o The authors need to provide information to how they created a database for proteomic searches.
o The authors should provide information on how they searched this manually curated database. Lines 502-503
o The authors should provide information to how many sequences were generated in this database and how the proteomic information matched against the transcriptome.
- The authors only used trinity to assemble the transcriptome. To get a more complete assembly and a better representation of the different toxins. https://doi.org/10.3390/toxins10060249
- How was bowtie 2 used to check the quality of the transcriptome? Line 434
For the proteomic analysis the authors present the relative abundance of the different annotated toxins. I have a few concerns about the analyses of the proteomics.
- How did the authors calculate the relative abundance of the proteomics? Since the authors used bands cut out of a gel for 11 different bands how did they quantify the abundance across the different bands? Can the authors add this information to the methods? I am fine with authors giving a list of the number of proteins identified from the proteomics but the abundance metrics I don’t think can be included with the current description of the methods.
- The proteins that were “uncharacterized” were manually identified used the same databases that were used in the original naming. Can the authors provide more information for how they manually annotated these uncharacterized sequences and how this is different from the general pipeline for the other proteins?
- The percentages in supplemental file 2 are not clear to me. How do these percentages add up across the different bands? Can the authors provide more detail to what these values are?
Lines 133-140 and lines 258-259: How were some proteins separated out to be listed as toxic proteins? Can the authors provide more information to how they separated the proteins into toxic and non-toxic components? I was confused when they stated that most of the proteins were toxic proteins in the discussion.
Not as familiar with Zymography analyses but when I looked zymography in other venom papers it appears that they have a molecular weight marker in the same gel. Can the authors clarify why there is what appears to be two gels put together on figure 7?
- Can the authors clarify why a negative control wasn’t included in this analysis?
- Can the authors highlight the positive control areas as well as the experimental ones in figure 7? What was the molecular weight of the positive control and did it align with the clear band in the gel?
Table 5: Can the authors display the results for the negative control as well in this table?
In the discussion the authors place a major emphasis on the prey capture ability of the venom and do not focus as much on the defensive potential of the venom. Can the authors provide more perspective the defensive aspect of the venom and tone down the discussion about the prey capture? I don’t think that the experiments done within this study can separate between whether the toxins are mainly functioning in a defensive or offensive role.
Minor points:
Remake the bar graphs so that the error bars are present on above and below the graph. Also add to the description of the bar graphs what the error bars represent (standard deviation, standard error?)
Line 12: Reword the first line of the abstract for clarity.
Line 14: “while avoiding the withdrawal of many animals, preserving ecosystem integrity” can the authors reword this part for clarity.
Line 28 and 34: “Phylum” does not need to be italicized
Line 34: Remove “also”
Line 37-39: Reword
Line 40-44: Split up this sentence
Line 48: Change exposition to exposure
Line 59: “difficulty-in some case- in” I am unsure what the authors are trying to say
Line 75-76: I am unsure what the authors are trying to say
Lines 80-82: Reword for clarity
Line 153: define “SD” with the first use
Line 220: What is X units of PLA2?
Define U/mg in the table caption
Line 295: remove “as reported in the results section”
Line 315: What is Xug/mL?
Line 317-319: Rephrase for clarity
Line 327: What do the authors mean by senile in this sentence?
Line 369 -372: Can the authors reword this sentence. I am unsure of what they are trying to convey.
Line 374-378: This comment is very confusing. There was not really any discussion on this topic. I think that the authors should either remove this section or add additional discussion in this paragraph.
Non-harmful animals is a subjective term that I don't think is very helpful. These animals can be harmful to their prey or to potential predators. If the authors want to talk about the effects on humans they should use the term not medically significant.
Line 580: It seems like the authors did not finish this sentence.
Why is the conclusion below the methods? Can this be moved to before the methods?
Line 647-649: Low toxicity does not mean it is helpful for drug design. Some of the current drugs derived from venom have very toxic effects. Can the authors provide citations for this argument and add more discussion on this?
Comments on the Quality of English Language
Comments to fix some grammar sentence structures are in the above comments.
Author Response
Reviewer 2
Added phrases addressing R2’s comments were highlighted in green in the text.
Comments from the reviewer and responses from the authors:
The authors present a massive amount of work describing the venom composition and function of venom from Velella velella. The authors assembled a previously published transcriptome and conducted a proteomic analysis of the venom. The authors then performed numerous experiments set to isolate and describe different enzymatic properties of the venom.
Major points:
The current description of the results of the transcriptome are very confusing. The authors do not present how many sequences that they assembled for toxin annotation or how many of those sequences matched with the proteome. The authors present the BUSCO scores of how many conserved genes were assembled with the trinity which help to understand the quality of the assembly but does not give any information to how the transcriptome was useful to the study.
- Could the authors provide more information to how the transcriptome was actually used and what information was pulled from this data?
- The authors need to provide information to how they created a database for proteomic searches.
- The authors should provide information on how they searched this manually curated database. Lines 502-503
- The authors should provide information to how many sequences were generated in this database and how the proteomic information matched against the transcriptome.
- The authors only used trinity to assemble the transcriptome. To get a more complete assembly and a better representation of the different toxins. https://doi.org/10.3390/toxins10060249
The authors thank the reviewer for the precise suggestions. The public set of SRA present on NCBI (SRX4941821), coming from a complete transcriptomic analysis of a floating colony of V. velella, was assembled using the Trinity software. Following the example of previous papers [Li et al., 2014 https://doi.org/10.1016/j.jprot.2014.04.011; Li et al., 2016 https://doi.org/10.1016/j.jprot.2016.07.023; Wang et al., 2018 https://doi.org/10.1021/acs.jproteome.8b00735; Liang et al., 2019 https://doi.org/10.1016/j.jprot.2019.103483], Trinity was the only software selected for the assembly, due to its widespread use and proven reliability. Even if other softwares, like the one reported in the paper cited by the reviewer, are able to provide a more in-depth isoform analysis, this kind of detailed information falls out the aim of the present study; however, future work may focus on a more comprehensive isoform investigation. As the Trinity outcome provides a set of nucleotide sequences (in our case, 234123 assembled sequences were obtained, as reported in S1), they had to be translated into protein sequences, to be used as a reference for the proteomic analysis. To achieve this, the Transdecoder software was used (see the Materials and Methods section). It allows to translate a database of nucleotide sequences into aminoacid sequences. Subsequently, the translated transcriptome will be used as a reference database for the mass spectrometry results.
Regarding the mass spectrometry analysis, the crude extract from the animals was first subjected to standard procedures in order to make it suitable for MS; then, the results obtained from the MS analysis were searched into the aminoacid sequences database previously obtained with the Transdecoder software: this procedure gave 783 different matches, corresponding to the proteins furtherly analyzed in the paper. To better clarify these concepts, a phrase was added in the methods section (line 450 and line 517).
- How was bowtie 2 used to check the quality of the transcriptome? Line 434
The authors acknowledge the reviewer’s observation. Bowtie2, in fact, is primarily used as an alignment tool, but it can be employed indirectly to assess the quality of an assembly by mapping the original reads back to the assembled sequences, thus gathering information on the completeness and accuracy of the assembly. Considering the percentage of original reads that align to the assembly, a high alignment rate means that the majority of the reads trace back to the assembly, thus indicating that the outcoming assembly is a good representation of the input data.
For the proteomic analysis the authors present the relative abundance of the different annotated toxins. I have a few concerns about the analyses of the proteomics.
- How did the authors calculate the relative abundance of the proteomics? Since the authors used bands cut out of a gel for 11 different bands how did they quantify the abundance across the different bands? Can the authors add this information to the methods? I am fine with authors giving a list of the number of proteins identified from the proteomics but the abundance metrics I don’t think can be included with the current description of the methods.
- The percentages in supplemental file 2 are not clear to me. How do these percentages add up across the different bands? Can the authors provide more detail to what these values are?
The aim of this work was primarily to qualitatively describe the protein content of the crude venom extract of V. velella. After electrophoresis, the excision points of the gel were oriented into following the protein bands pattern observed in the SDS-PAGE, but including also zones of the lane where the presence of proteins was not visibly evident. This allowed to divide the entire lane into 11 sections that were called “bands” for the sake of simplicity. In each band, a certain amount of different proteins was found, divided by their molecular weight range. For each molecular weight range, different functional groups of proteins were found, as reported in Supplementary 2: for instance, in band 1 were identified 95 different proteins, of which the 22.11% belonged to the group “Structural Components and Extracellular Matrix”, the 16.84% belonged to the group “Membrane Transport and Channel Activities”, and so on (considering nproteins = 95 as the 100% of the proteins for Band 1). The same approach was used to calculate the qualitative composition of the total protein content in the crude extract, in this case considering nproteins = 783 (which is the number of different proteins identified after the MS analysis) as the 100%, and considering all the proteins belonging to a functional group, regardless of their molecular weight. A quantitative analysis, for instance using a densitometry approach, wasn’t part of the aims of this paper, but it could definitely be considered for future works. Line 129 and the caption of Figure 2 were rephrased to better highlight the qualitative nature of the analysis.
- The proteins that were “uncharacterized” were manually identified used the same databases that were used in the original naming. Can the authors provide more information for how they manually annotated these uncharacterized sequences and how this is different from the general pipeline for the other proteins?
The first annotation was performed using the BLAST software, which is primarily a sequence alignment tool, used to compare queried proteins (in our case, the MS results that found a match within the translated V. velella transcriptome) against a database of known sequences (in our case, an invertebrate’s proteins database) to find regions of similarity, thus identifying homologous sequences and giving an identity to an unknown protein sequence. This analysis was performed at the same time on the entire set of proteins that we wanted to identify. However, query sequences may not find a correspondence, especially in the case of non-model species. In this case, proteins might be distantly related to known proteins in terms of aminoacid sequence, but they may retain a 3D structure (and, consequently, a homologous function) similar to better known proteins. For this reason, the proteins that came out as “uncharacterized” after the first attempt of annotation were subjected to a more in-depth analysis in order to identify their homologues and their functional category. To achieve this, each one of the uncharacterized sequences was singularly subjected to a structural prediction analysis, using Swiss Model and Phyre2 (both of which provide structural prediction based on homology modelling), and the SMART tool (which primarily annotates conserved domains). This combined approach allowed us to obtain a more precise understanding of the previously unknown proteins’ function. A phrase was added at the end of paragraph 4.6.3. to better clarify the difference in the two pipelines.
Lines 133-140 and lines 258-259: How were some proteins separated out to be listed as toxic proteins? Can the authors provide more information to how they separated the proteins into toxic and non-toxic components? I was confused when they stated that most of the proteins were toxic proteins in the discussion.
The authors thank the reviewer for pointing out a topic that could be misunderstood. To understand how the potentially toxic components were separated from the other components, is important to notice that a toxic activity, broadly speaking, can be displayed not only by molecules that are specifically referred to as “toxins” (like pore-forming toxins, neurotoxins, and so on), but also –more or less directly– by other molecules, like hydrolytic enzymes, lectins, and proteins related to immune responses. In Figure 2, the black slice of the pie-chart (1.79% of the total protein content), is, indeed, referred to “true” toxins; however, basing on literature, other protein categories represented in Figure 2 were took into consideration for having potentially toxic activity and furtherly categorized (together with the “true” toxins”) in Table 3. The potentially toxic elements were selected especially considering the following cited papers:
- [15] D’Ambra, I.; Lauritano, C. A Review of Toxins from Cnidaria. Marine Drugs 2020, 18, 507, doi:3390/md18100507. In Table 2, D’Ambra and Lauritano report the different types of toxic molecules identified in cnidarians.
- [31] Jaimes-Becerra, A.; Chung, R.; Morandini, A.C.; Weston, A.J.; Padilla, G.; Gacesa, R.; Ward, M.; Long, P.F.; Marques, A.C. Comparative Proteomics Reveals Recruitment Patterns of Some Protein Families in the Venoms of Cnidaria. Toxicon 2017, 137, 19–26, doi:1016/j.toxicon.2017.07.012. In Table 1, Jaimes-Becerra et al. report the predicted venom proteomes of potential toxins isolated from the nematocysts of different species of cnidarians.
- [32] Doonan, L.B.; Lynham, S.; Quinlan, C.; Ibiji, S.C.; Winter, C.E.; Padilla, G.; Jaimes-Becerra, A.; Morandini, A.C.; Marques, A.C.; Long, P.F. Venom Composition Does Not Vary Greatly Between Different Nematocyst Types Isolated from the Primary Tentacles of Olindias sambaquiensis (Cnidaria: Hydrozoa). The Biological Bulletin 2019, 237, 26–35, doi:1086/705113. In Table 1, Doonan et al. report the potential toxic activities predicted in the venom of the hydrozoan Olindias sambaquiensis.
- [33] Jaimes-Becerra, A.; Gacesa, R.; Doonan, L.B.; Hartigan, A.; Marques, A.C.; Okamura, B.; Long, P.F. “Beyond Primary Sequence”—Proteomic Data Reveal Complex Toxins in Cnidarian Venoms. Integrative and Comparative Biology 2019, 59, 777–785, doi:1093/icb/icz106. In Table 1, Jaimes-Becerra et al. listed potential venom toxins found with the proteomic analysis of three different species of cnidarians.
Considering these data, we extrapolated from our results all the proteins that could fall into the cited categories, simultaneously excluding those proteins that clearly belonged into certainly non-toxic categories (such as collagens, actin, myosin, and so on), and that –as reported in Figure 2– constituted the greatest fraction of our extract.
Regarding the discussion, the intention wasn’t to state that most of the protein are toxic, but that, among the potentially toxic elements, hydrolases are the most represented group. To avoid misunderstandings, the sentence at lines 264-267 has been reworded and made clearer.
Not as familiar with Zymography analyses but when I looked zymography in other venom papers it appears that they have a molecular weight marker in the same gel. Can the authors clarify why there is what appears to be two gels put together on figure 7?
- Can the authors clarify why a negative control wasn’t included in this analysis?
- Can the authors highlight the positive control areas as well as the experimental ones in figure 7? What was the molecular weight of the positive control and did it align with the clear band in the gel?
The authors agree that Figure 7 may not be easy to interpret and thank the reviewer for the suggestions. Panels A and B represent two gels embedded with different protease substrates (casein in Panel A and gelatine in Panel B). During the experiment, the standards and samples were run simultaneously; however, after electrophoresis, the standards were separated from the rest of the gel because they were pre-stained. This precaution was taken because the protein substrates embedded in the gels cause overall staining, which could reduce the visibility of the pre-stained standards against the dark background. For visualization purposes, the portion of the gel containing the standards was reassembled with the stained portion, as evident from the complementary alignment of the two pieces.
A negative control was indeed included in the original experiment; however, it was omitted from the final image due to its lack of visible results. The negative control lane did not display any bands, which was expected, and including it in the figure would have required enlarging the image, thereby reducing the resolution and compromising the clarity of the positive control and experimental samples; for these reasons, we prioritized showing the comparison between the positive control and the crude extract samples.
The positive control used in this experiment is a commercial mix of Streptomyces griseus proteases, which results in multiple bands due to the presence of different proteolytic enzymes. Therefore, the exact molecular weight of the expected bands cannot be specified. However, the observed bands in the positive control lane do confirm the presence of proteolytic activity and serve as a valid comparison for the enzymatic activity observed in the experimental lanes. To improve clarity, we revised Figure 7 and highlighted the areas corresponding to the proteolytic activity in the positive control; nonetheless, due to other reviewer suggestions, we decided to move the figure from the main text to another Supplementary file (S1), together with the figure representing the DNAse activity.
Table 5: Can the authors display the results for the negative control as well in this table?
We did not include the results for the negative control in Table 5 because the absorbance values for the negative control were below the detection limit of the assay, resulting in negative absorbance readings. Since negative absorbance values do not provide meaningful or interpretable data within the context of this experiment, we opted to exclude them to avoid potential confusion. However, the presentation of results has been modified according to the suggestions of other reviewers, therefore this table is no longer present.
In the discussion the authors place a major emphasis on the prey capture ability of the venom and do not focus as much on the defensive potential of the venom. Can the authors provide more perspective the defensive aspect of the venom and tone down the discussion about the prey capture? I don’t think that the experiments done within this study can separate between whether the toxins are mainly functioning in a defensive or offensive role.
The authors thank the reviewer for this comment. In fact, with the information available to date, we cannot state the actual biological role of the molecules we found in V. velella. Hence, we revised this aspect of the discussion as suggested.
Minor points:
Remake the bar graphs so that the error bars are present on above and below the graph. Also add to the description of the bar graphs what the error bars represent (standard deviation, standard error?)
We thank the reviewer for the insightful observation and have decided to revise the presentation of our results, incorporating feedback from other reviewers as well. Specifically, we have replaced the graphs depicting enzyme activities with a table that summarizes the activity detected for each enzyme. This format allows for a clearer and more concise presentation of the data. Given that proteases represent the largest group of enzymes identified in our study, and their activity was further analyzed using specific inhibitors, we have chosen to maintain a separate presentation for them. Consequently, no graphs are included in the revised manuscript, and all data are now presented in tabular form.
Line 12: Reword the first line of the abstract for clarity.
We revised this part of the abstract and made it clearer.
Line 14: “while avoiding the withdrawal of many animals, preserving ecosystem integrity” can the authors reword this part for clarity.
This part was reworded according to the suggestion.
Line 28 and 34: “Phylum” does not need to be italicized
The word “phylum” was changed where needed.
Line 34: Remove “also”
The modification has been done.
Line 37-39: Reword
This sentence has been revised for more clarity.
Line 40-44: Split up this sentence
The sentence has been split up according to the suggestion.
Line 48: Change exposition to exposure
The modification has been done.
Line 59: “difficulty-in some case- in” I am unsure what the authors are trying to say
This was changed in: “[…] also accounting how challenging it is to obtain enough fresh specimens […]”. The authors meant to say that the proteomic approach allows to study the complexity of a venom without withdrawing many animals from the environment, also considering that is not always easy to collect fresh specimens.
Line 75-76: I am unsure what the authors are trying to say
Here, the authors are describing the anatomical structure of Velella velella. The sentence was reworded in this way for more clarity: “[…] these specialized polyps are enclosed by an additional ring of dactylozooids, which are involved in prey capture and contain the highest number of nematocysts, being mainly of the eurytele and stenotele types”
Lines 80-82: Reword for clarity
The sentence has been modified according to the suggestion.
Line 153: define “SD” with the first use
“SD” has been extended to “standard deviation” for the first use.
Line 220: What is X units of PLA2?
The presentation of results has been changed; however, this typo has been corrected in Table 5.
Define U/mg in the table caption
“U/mg”, i. e., enzyme units/mg of extract, has been extended in the caption of Table 5.
Line 295: remove “as reported in the results section”
The wording has been removed according to the suggestion.
Line 315: What is Xug/mL?
This typo was referred to the amount of chitinase activity found in the extract of V. velella. It has been modified to “0.070 ± 0.001 (SD) U/mg”.
Line 317-319: Rephrase for clarity
The sentence has been reworded as suggested.
Line 327: What do the authors mean by senile in this sentence?
The word “senile” has been replaced with “senescent”. Senescent symbionts are symbiotic organisms that have aged and experienced functional decline, gradually losing their ability to contribute to the host in mutualistic interactions. As these symbionts become less beneficial, the host may either expel them or use them as a nutrient source.
Line 369-372: Can the authors reword this sentence. I am unsure of what they are trying to convey.
This sentence has been modified and made clear and more flowing, thanks to the reviewer suggestions.
Line 374-378: This comment is very confusing. There was not really any discussion on this topic. I think that the authors should either remove this section or add additional discussion in this paragraph. Non-harmful animals is a subjective term that I don't think is very helpful. These animals can be harmful to their prey or to potential predators. If the authors want to talk about the effects on humans they should use the term not medically significant.
Acknowledging the potential confusion that the term “non-harmful” could generate, and recognizing that this wording is incorrect, we completely revised the last paragraph of the discussion section (also following the suggestions of other reviewers) and changed “non-harmful” with “not medically relevant”.
Line 580: It seems like the authors did not finish this sentence?
We thank the reviewer for the correction and finished the sentence accordingly.
Why is the conclusion below the methods? Can this be moved to before the methods?
In the template provided by the Marine Drugs journal, the Conclusions section (even if not mandatory) has to be put at the end of the article, below the methods.
Line 647-649: Low toxicity does not mean it is helpful for drug design. Some of the current drugs derived from venom have very toxic effects. Can the authors provide citations for this argument and add more discussion on this?
The authors acknowledge the concern of the reviewer about this topic. Venoms are typically composed of toxic molecules, such as ion channel blockers, accompanied by numerous other components. Many of these additional components—found even in venoms from lethal animals—are not inherently toxic on their own but act as spreading factors, enhancing the activity of the primary toxins. Some enzymes, like hyaluronidases and phospholipases, or non-enzymatic molecules like lectins, also possess significant pharmacological potential and deserve investigation as biologically active compounds. In the crude venom extract of V. velella, we identified both spreading factors and molecules homologous to well-known toxins, such as Alpha-elapitoxin and Kappa-stichotoxin. While a toxin's harmfulness depends on factors like its ability to bind specific targets (e.g., ion channels), the route of exposure, and its concentration, it is safe to conclude that V. velella’s venom does not produce medically relevant effects on humans. This could be due to its toxins being slightly different from those in animals more dangerous to humans, preventing the triggering of harmful physiological cascades. This lower toxicity could actually be advantageous in drug discovery. While many drugs, such as Ziconotide, are derived from highly toxic venom compounds, their extreme toxicity often complicates clinical trials and requires chemical modifications to reduce harmful effects​ [de Castro Figueiredo Bordon et al., 2020 https://doi.org/10.3389/fphar.2020.01132]. In contrast, low- or non-toxic molecules, such as those in V. velella, may require fewer adjustments, making them more suitable for developing into drugs with minimal side effects. For instance, compounds like crotamine have demonstrated cytotoxic activity in specific contexts without causing harm to healthy tissues​ [Oliveira et al., 2022 https://doi.org/10.1038/s41570-022-00393-7]). Given these points, we have revised both the discussion and conclusion to better explain this concept.
Reviewer 3 Report
Comments and Suggestions for Authors
This article reports on the proteomic and biochemical characterization of a crude nematocyst extract from the hydrozoan Velella velella, which is not dangerous to humans.
The proteomic analysis of the extract, combined with the analysis of the transcriptome of this species, also conducted by the authors, led to the identification of 783 proteins, which the authors categorized into structural components, enzymes, and potential toxins. The authors analyzed, using several modeling softwares, 116 out of 783 sequences that didn’t match the BLAST database in order to categorize them, and the biochemical assays conducted by them confirmed the presence of proteases, phospholipases, hyaluronidases, DNases, and chitinases.
The authors’ findings open the way for future studies related to the pharmacological application of some components of the venom of this species.
This reviewer considers that the identified sequences should be explicitly presented in the article; since they are very numerous, they should at least be included in the "Supplementary Material".

Comments on the Quality of English Language
The article is very well written; only some details should be addressed. In addition, there are several inconsistencies in the references that the authors should correct according to the current requirements of Marine Drugs (Please, see the uploaded file: marinedrugs-3175750-peer-review-v1_rev).
Author Response
Reviewer 3
Added phrases addressing R3’s comments were highlighted in pink in the text.
Reviewer comments:
This article reports on the proteomic and biochemical characterization of a crude nematocyst extract from the hydrozoan Velella velella, which is not dangerous to humans.
The proteomic analysis of the extract, combined with the analysis of the transcriptome of this species, also conducted by the authors, led to the identification of 783 proteins, which the authors categorized into structural components, enzymes, and potential toxins. The authors analyzed, using several modeling softwares, 116 out of 783 sequences that didn’t match the BLAST database in order to categorize them, and the biochemical assays conducted by them confirmed the presence of proteases, phospholipases, hyaluronidases, DNases, and chitinases.
The authors’ findings open the way for future studies related to the pharmacological application of some components of the venom of this species.
This reviewer considers that the identified sequences should be explicitly presented in the article; since they are very numerous, they should at least be included in the "Supplementary Material".
Comments on the Quality of English Language
The article is very well written; only some details should be addressed. In addition, there are several inconsistencies in the references that the authors should correct according to the current requirements of Marine Drugs (Please, see the uploaded file: marinedrugs-3175750-peer-review-v1_rev).
Response from the authors:
The authors thank the reviewer for its valuable feedback. We revised the manuscript and modified Figure 2 according to his suggestions, and were thus able to improve the overall clarity and precision. Below, we provide our responses and clarifications to specific points raised:
- Table 1 - GO Terms: we weren’t able to furtherly disrupt the Table, as multiple GO Terms can be assigned to a single protein, as they describe different aspects of a protein’s function, location, or processes in which it is involved. For this reason, we thought that it should be better maintain wider categories into which multiple GO Terms could fit.
- Line 289 (discussion): we acknowledge the potential confusion caused by the wording and have revised the text for clarity. We are specifically referring to gelatinase, which can digest gelatin but not intact collagen. Our intention was to suggest the hypothesis that an enzyme with collagenolytic activity may be present, which could break down intact collagen and provide a substrate for the gelatinases to act upon.
- Line 430 (methods): In bioinformatics pipelines, even if not strictly necessary, it is standard practice to perform multiple quality checks on raw data. Initially, we used FastQC to assess the quality of the reads. Following this, a first round of trimming with Trimmomatic was conducted to remove reads that were too short. The trimmed data was then re-evaluated with FastQC to ensure the trimming process was successful. Finally, an additional step was performed to remove any remaining adapter sequences. These steps ensure cleaner sequencing data for reliable downstream analyses.
Reviewer 4 Report
Comments and Suggestions for Authors
The article titled “Proteomic analysis and biochemical characterization of the 2 nematocyst extract of the hydrozoan Velella velella” is devoted to the study of the marine animal extract composition. The authors used a comprehensive approach, obtained transcriptomic and proteomic data, and also presented some biochemical characteristics of the extract. However, despite the use of modern methods, the authors were unable to fully link these data (it is not clear why the transcripts were obtained), and this makes the article appear incomplete.
Major notes
The introduction is written very broadly and does not give any idea of the article purpose. I recommend expanding this part and introducing already known data to better represent the novelty and depth of research in this field.
I would strongly recommend reviewing the obtained results, selecting interesting findings from the general pool with following a little deeper into their analysis. Then the article will be interesting to researchers who study certain compounds, toxins, for example, or enzymes, or other. There is a lack of "live" material: the ratio (percentage), which, by the way, depends on sample preparation (!), as well as rough transcript data - this is in the databases, but the purpose of a scientific article is to solve (or possible ways to solve) some problem, as it seems to me. The article (and the supplementary) lack evidence of MS data, for example, but it provides detailed information (working points) for biochemical studies. It is necessary to expand the results of the transcript study, MS data, and shorten the description of the activity determination results. Some comments on the presentation of the results are given below.
Lines 90, 91, 153, 155 and others (especially Table 4): 0.1 ± 0.04 should be replaced 0.10 ± 0.04; 306.66 ± 11 should be replaced 306.66 ± 11.xx (??). Please bring all values with errors into compliance. The value cannot be measured more accurately (up to the second decimal) than the error (whole value).
Line 99. Figure 1 does not have any semantic load. The concentrations are indicated in the text, this is quite sufficient.
Line 102. It is not yet clear why the authors need this result. I recommend replacing “Transcriptome assembly” with “Transcriptome analysis” and expanding this part.
Line 108. I recommend starting with the electrophoresis results that you have in your methods. Then provide the MS data, and only then characterize the proteins.
Line 163. Table 4 does not have any semantic load. There is no point in giving the optical density values. The results will look better as a percentage of the control per unit of protein and their mention in the text will be enough. If we talk about the table, then I would recommend making a general table for all types of activity. Such a table will be useful.
Lines 170-173. The conclusion made by the authors based on only one concentration tested does not seem quite logical. If the authors want to confirm the proteomic data on the presence of collagenases, they would have to take a concentration (preferably several concentrations) so that this activity could be noticed (or change the method). It is strange to show a negative result in this regard. The graph is absolutely uninformative. It should either be in the appendix, and the results described as text data.
Line 191. The drawing is very "working". Only for the appendix.
Line 407. Scheme 1 is not a modified scheme by the authors, but demonstrates a normal routine process. It is not clear why it is shown.
Line 469. It is a result. Replace it, please, in Result
Line 517. The authors study common routine activities. In this case, it is necessary to use well-proven methods. And then the text should be shortened, obvious details removed and a reference should be made to a valid method (This also applies to electrophoresis in MS).
After the article has been reworked, the discussion should also change significantly.
Author Response
Reviewer 4
Added phrases addressing R4’s comments were highlighted in light blue in the text.
The article titled “Proteomic analysis and biochemical characterization of the 2 nematocyst extract of the hydrozoan Velella velella” is devoted to the study of the marine animal extract composition. The authors used a comprehensive approach, obtained transcriptomic and proteomic data, and also presented some biochemical characteristics of the extract. However, despite the use of modern methods, the authors were unable to fully link these data (it is not clear why the transcripts were obtained), and this makes the article appear incomplete.
Reviewer comments and responses from the authors:
Major notes:
The introduction is written very broadly and does not give any idea of the article purpose. I recommend expanding this part and introducing already known data to better represent the novelty and depth of research in this field.
The authors thank the reviewer for this valuable suggestion. We have expanded the introduction to provide a more focused overview of the study's purpose and its context within the existing literature.
There is a lack of "live" material: the ratio (percentage), which, by the way, depends on sample preparation (!), as well as rough transcript data - this is in the databases, but the purpose of a scientific article is to solve (or possible ways to solve) some problem, as it seems to me. The article (and the supplementary) lack evidence of MS data, for example, but it provides detailed information (working points) for biochemical studies. It is necessary to expand the results of the transcript study, MS data, and shorten the description of the activity determination results. Some comments on the presentation of the results are given below.
The authors acknowledge the reviewer’s recommendations and have expanded the manuscript to include more detailed evidence of the MS data and transcriptome analysis. Additionally, we have added both the MS data and the sequences of the identified proteins in the supplementary materials to provide a comprehensive overview.
Lines 90, 91, 153, 155 and others (especially Table 4): 0.1 ± 0.04 should be replaced 0.10 ± 0.04; 306.66 ± 11 should be replaced 306.66 ± 11.xx (??). Please bring all values with errors into compliance. The value cannot be measured more accurately (up to the second decimal) than the error (whole value).
The authors thank the reviewer for the clarification. We have revised the manuscript to ensure consistency in the presentation of the values and their respective errors and have uniformly adjusted all the results to two decimal places (when possible) to maintain clarity and coherence throughout the text.
Line 99. Figure 1 does not have any semantic load. The concentrations are indicated in the text, this is quite sufficient.
Acknowledging the comment of the reviewer, we removed this figure from the manuscript and put it into Supplementary File 1.
Line 163. Table 4 does not have any semantic load. There is no point in giving the optical density values. The results will look better as a percentage of the control per unit of protein and their mention in the text will be enough. If we talk about the table, then I would recommend making a general table for all types of activity. Such a table will be useful.
According to the reviewer’s suggestion, we modified Table 4 removing the OD values and reported all enzymatic activity into into one single table (Table 5). Table 4 was maintained, without the OD values, because the evaluation of the enzymatic activity of proteases includes different conditions if compared to the others (i.e., the presence of specific inhibitors) and we think it’s clearer if kept separated from other activities.
Lines 170-173. The conclusion made by the authors based on only one concentration tested does not seem quite logical. If the authors want to confirm the proteomic data on the presence of collagenases, they would have to take a concentration (preferably several concentrations) so that this activity could be noticed (or change the method). It is strange to show a negative result in this regard. The graph is absolutely uninformative. It should either be in the appendix, and the results described as text data.
For the collagenase assay, we used the highest possible concentration of the extract, which was used as well for the other assays. Increasing the concentration any further was not feasible because we observed that higher concentrations caused the proteins to precipitate; lower concentrations wouldn’t give visible results. However, according to the reviewer’s suggestions, we removed the graph and reported it into S1.
Line 191. The drawing is very "working". Only for the appendix.
The figure has been removed and put into S1.
Line 108. I recommend starting with the electrophoresis results that you have in your methods. Then provide the MS data, and only then characterize the proteins.
Accordingly to the reviewer’s suggestions, we have restructured the exposition of the results. These changes enhance the overall flow of the text, thus presenting data in a more meaningful and concise manner.
Line 469. It is a result. Replace it, please, in Result
The SDS-PAGE of the crude venom extract has been added to the results.
Line 102. It is not yet clear why the authors need this result. I recommend replacing “Transcriptome assembly” with “Transcriptome analysis” and expanding this part.
We understand the need for clarification. We have replaced “Transcriptome assembly” with “Transcriptome assembly and analysis” to better reflect the content of the section. Additionally, we have expanded this part to clarify that the transcriptome was used as a species-specific reference database for the proteomic analysis. This result was needed because it was essential for accurately mapping and identifying the proteins found in the nematocyst extract through the proteomics analysis.
Line 407. Scheme 1 is not a modified scheme by the authors, but demonstrates a normal routine process. It is not clear why it is shown.
Given the various protocols available for extracting nematocyst content, each potentially yielding different results depending on the method used, we chose to report our specific extraction method in detail, aiming to provide clarity for other researchers who may wish to replicate our approach and ensure consistency in their own studies.
Line 517. The authors study common routine activities. In this case, it is necessary to use well-proven methods. And then the text should be shortened, obvious details removed and a reference should be made to a valid method (This also applies to electrophoresis in MS).
We appreciate the reviewer’s comments and understand the concern about including routine methodologies. Some protocols, like those for zymograms, may be very similar between each other, but we noticed that even a slight modification can affect the obtainment of a good result. In fact, we encountered challenges in identifying and adapting methods that best suited our specific needs. For instance, the chitinase assay was optimized based on various methods from previous studies. We believe that providing a detailed description of these methods will benefit future researchers working in this area, as it allows for reproducibility and highlights how the choice of extraction methods can influence the results. For these reasons, we included a thorough description of our methodologies.
I would strongly recommend reviewing the obtained results, selecting interesting findings from the general pool with following a little deeper into their analysis. Then the article will be interesting to researchers who study certain compounds, toxins, for example, or enzymes, or other. After the article has been reworked, the discussion should also change significantly.
The authors thank the reviewer for the valuable feedback. The manuscript has been revised according to the suggestions, focusing on the most interesting findings among the variety of compounds identified within the venom, and the discussion has been implemented as well.
Round 2
Reviewer 2 Report
Comments and Suggestions for Authors
The authors provided a lot of support and added clarification for the paper and some of the methods. I appreciate all of the comments that the authors gave.
I have a few things that I still feel like should be addressed. The authors in the their response state that "Trinity was the only software selected for the assembly, due to its widespread use and proven reliability. Even if other softwares, like the one reported in the paper cited by the reviewer, are able to provide a more in-depth isoform analysis, this kind of detailed information falls out the aim of the present study; however, future work may focus on a more comprehensive isoform investigation" The paper that I cited talks about how using a single method can limit the diversity that is found within a de novo assembled transcriptome among families of proteins and not just related to isoforms of the same proteins they identified. Can the authors provide more information on their choice of methods?
On the same section about the transcriptome the authors state that they used transdecoder to translate the assembled transcriptome to be used with the proteome but do not give any other information about how this program was used. Did the authors use all available reading frames or only the first reading frame? Were all sections of the transcripts used in the database or only a subset of the translated sections (was there any filtering based on length, start or stop codons, ect)?
On the results section for the transcriptome analysis on lines 113 -114 the authors present that the transcriptome consisted of 76% complete sequences. These values are only applicable to the BUSCO genes that were found in the QC of the transcriptome. This section needs to have this sentence reworded for clarification but it would also be made better by reporting either the actual number of sequences that were assembled or the number of sequences that were assembled that matched with parts of the proteome as is given on lines 276-277.
For the qualitative analysis of the proteins, I now understand how the authors made both figure 2 and 3. I think that similar wording needs to be placed in the main manuscript. On lines 158 and 159 the authors use "percentage composition". This to me still implies that they are looking at the relative abundance of these protein families in the venom based on the MS counts and not based on how many different proteins were identified. Can the authors please clarify this better in the manuscript.
On table 5, the authors have the titles as enzymatic activity and quantity of enzyme in crude extract. I think that these titles are confusing. Maybe it should be Enzymatic Test in the first column and Enzymatic Activity in the crude venom extract in the second column.
I think that the authors did a great job restructuring and rewriting their discussion and conclusion based on my comments and the other reviewers!
